# Governance of Heritable Human Gene Editing World-Wide and Beyond

**DOI:** 10.3390/ijerph19116739

**Published:** 2022-05-31

**Authors:** Yang Xue, Lijun Shang

**Affiliations:** 1Law School, Tianjin University, Tianjin 300072, China; xueyang@tju.edu.cn; 2Center for Biosafety Research and Strategy, Tianjin University, Tianjin 300072, China; 3School of Human Sciences, London Metropolitan University, London N7 8DB, UK; 4Biological Security Center, London Metropolitan University, London N7 8DB, UK

**Keywords:** heritable human gene editing, medical practice, regulatory science, codes and education, regulation, democratic deliberation, global governance

## Abstract

To date, the controversy surrounding the unknown risks and consequences of heritable genome editing has grown, with such work raising biosafety and ethical concerns for future generations. However, the current guideline of global governance is limited. In the context of the new framework for the governance of human genome editing developed by the World Health Organization (WHO) committee, this paper presents further analysis by highlighting predicaments of governance on germline engineering that merit the most attention: (1) building a scientific culture informed by a broader set of values and considerations in the internal scientific community at large, such as codes of ethics, and education, in addition to awareness-raising measures; and (2) reflecting on and institutionalizing policies in grassroots practice according to local conditions in external governance, such as the experimentalist governance, which is a multi-layered model of governance that establishes an open-ended framework from the top and offers stakeholders the freedom of discussion. The key to achieving these goals is more democratic deliberation between the public and the inclusive engagement of the global scientific community, which has been extensively used in the Biological and Toxin Weapons Convention (BTWC). On a global scale, we believe that practicing heritable human genome editing in accordance with the WHO and BTWC appears to be a good choice.

## 1. Introduction

Human genome editing has great potential to improve human health and medicine [1]. These techniques can be used on somatic cells (non-heritable), germline cells (not for reproduction), and germline cells (for reproduction) [2]. As the most rapidly emerging technology in recent years, human genome editing is seeking to modify the genes of living organisms to improve our understanding of gene function and advance potential therapeutic applications to correct genetic abnormalities. Its potential benefits include new avenues to treat infertility new ways to promote disease resistance, contribution to vaccine development, and enhanced knowledge of human biology in general. For example, human genome editing is already widely used in basic research and is in the early stages of development and trials for clinical applications that involve non-heritable (somatic) cells. Germline genome editing, in contrast, is contentious because genetic changes could be inherited by the next generation. The concept of altering the human germline in embryos for clinical purposes has been debated over many years from many different perspectives [3] and has been viewed almost universally as a line that should not be crossed. Before heritable germline editing can fulfill the proper risk and benefit standards for clinical trials, much more research is required. In November 2018, He Jiankui, a biophysicist working at the Southern University of Science and Technology in China, reported using edited embryos to start pregnancies that resulted in the birth of at least two children. The modification of human embryonic cells used by He Jiankui was based on *CCR5*, a target used to stop HIV from damaging the body’s immune system and to prevent HIV transmission to prospective persons where the male progenitor was HIV-positive [4]. Given the availability of other means for preventing vertical transmission of HIV, his argument for modifying this gene had little to no clinical justification [5,6]. Consequently, he was found guilty by a Chinese Court of illegal medical practice and sentenced to three years in prison [7]. This case has stirred deep concerns all over the world and underlined the urgency of questions about “responsible translational pathways” for human germline gene editing [8,9].

The scientific community has begun to consider these issues, rethinking and debating heritable genome editing of early embryos, eggs, sperm, or precursor cells in addition to its implications in the foreseeable future. Difficult questions arise regarding the latter—for example, to what extent is human germline engineering inevitable? What are the major outstanding technical barriers to achieving germline alteration for human clinical application? What are the individual health risks and the societal risks of germline engineering? Which is the optimal approach for over-sight: full international ban, temporary moratorium, regulation, or laissez-faire approaches? Some general actions have been agreed upon, such as encouraging public input and implementing regulations on preclinical and clinical research in human heritable genome editing, particularly in those technological engagements where unjustified, unpredictable, and foreseeable characteristics bring strong arguments. In the end, according to the report from the National Academies of Sciences, Engineering, and Medicine, “Public acceptance may change as the benefits of genome editing emerge for disease prevention. Eventually, as we move from research to the clinic, the collective sum of individual decisions could constitute a de facto policy” [10].

However, while debate has seen a focus on the clinical, scientific, and technical aspects of the work, the effectiveness of such debate in guiding global governance is limited. Great efforts should be made to ensure that the advanced technology and research fit into society, and that the public does not remain passive and deferential in the face of specialized scientific and technological knowledge. Moreover, it is necessary to analyze the predicament of governance in germline engineering with its rapid development and explore approaches for establishing a robust, credible, and lasting regulatory regime.

This paper is therefore designed to address the above issues by beginning to analyze the current public’s understanding of the legitimacy of heritable human gene editing advance and outstanding issues on strong arguments against engaging in this technology. Crucially, this paper will answer those questions which can be used to serve as a guideline for multiple stakeholders who are engaged in deliberation and illustrate the characteristics of internal autonomy and external governance which have been created by the translation and application of the technological results of heritable human genome editing. This paper will then use the regulatory updates of heritable genome editing in China and the US as specific examples to make recommendations on appropriate human genome editing governance mechanisms for other counties. This paper proposes to address two ultimate questions: (1) whether the internal scientific community could build a scientific culture informed by a broader set of values and considerations at all, and (2) whether external governance could reflect on and institutionalize policies in grassroots practice according to local conditions.

## 2. The Legitimacy of Biotechnological Advance

### 2.1. The Substance of “Advance” in Biotechnology

With the advancement of science and technology, the public and policymakers are increasingly involved in the discussion, particularly in science and technology policies [11]. This heavy involvement of the public and policymaker in deputes that are originally limited within the scientific communities might be due to the following reasons: first, scientific and technological achievements not only bring dividends to social life but also indirectly intensifies their direct interaction with non-expert groups; second, the public increasingly values the right to know and the right to speak; third, while promoting theoretical and technological advances, scientists are constantly predicting the risks and effects that technological advancement may bring. Although the action above is indeed due to scientists’ professional ethics, it has intensified the disintegration of their professional authority and status. The controversy surrounding the unknown risks and consequences of human germline and heritable genome editing has grown in breadth and intensity [12,13,14]. It not only demonstrates public dissatisfaction with current policy considerations [15], but also expresses publicly and scientifically different viewpoints on whether such heritable genome editing is needed or not [16,17,18]. Scientists strive to cure disease and improve environmental health, and yet the dominant values ingrained in scientists center on the virtues of independence, ambition, and objectivity, which is a grossly inadequate set of skills with which to support a mission of advancing society [19]. Ogburn observed that as technology advanced, people’s habits, thoughts, and social arrangements fell behind, resulting in the decisive influence of technology on society [20]. The profound implication of this dynamic is that scientific endeavors have greatly outstripped the social and cultural acceptance to which they belong. Adding to these concerns, the regulations in many countries have not kept pace with science.

### 2.2. Changing in Public Attitude

Heritable human genome editing involves inquiring about human origins, traits, and health, in addition to addressing technical theoretical issues in medical treatment. The former trend has become increasingly evident, particularly in response to certain strong social demands. From the initial unconditional acceptance of gene editing advances, the public is now casting their doubts. For example, the public believes that scientists often take internal decisions to completely change the life of human society without fully discussing and forming an agreement with them, and the public feel frustrated as their involvement in the debates is normally after the decision has been made [21]. These changes are positive and worth recognition. The reasons for these changes could originate from the following:

#### 2.2.1. Biotechnology on the Changing from Dominating Nature to Dominating Subjects

The reason why biotechnology research was unconditionally accepted by the public at the initial stage is that science and technology often claim to be servants of society [22]. However, when the advance of biotechnology is likely to infringe upon the basic rights of human beings, such as safety concerns [23,24,25], informed consent [14], challenges to human dignity [26], and the potential for permanent negative impact on future generations, including its abuse for eugenics or enhancement (the parental pursuit of specific traits for non-medical reasons) [27,28], the scope involved turns into the problem of “man and man” and “man and society”. This is because biotechnology’s original goal of ruling nature has shifted to technological dominance over human subjects, regardless of whether the reality has resulted in a significant infringement or is speculation about the future. As a result of the aforementioned changes, biotechnology may face resistance from social culture, ethics, and morality.

For example, the accessibility of a powerful genetic engineering tool has already led to ethical challenges with Chinese scientist He Jiankui’s engineering of human embryonic genomes [29]. He Jiankui’s work violated a longstanding norm prohibiting genetic modification of the human germline, where these modifications may be passed on to future generations [30]. Even if gene editing technology is safe, effective, and affordable, it also raises relevant concerns such as fairness, social justice, and non-discrimination, in addition to potential disregard for the moral respect owed to vulnerable groups [31]. It highlighted a symptom of a broader scientific cultural crisis: a growing divide between the values upheld by the scientific community and the mission of science itself. Therefore, in essence, the deterioration of the relationship between “man and man” and “man and society” is hidden behind the deterioration of biotechnology in the relationship between “man and nature”. We must reflect on how our research fits into the legislations and ethics which require not just our intellects, but also our emotions. It is worrisome that in the pursuit of objectivity, science seems to lose its original purpose [19].

#### 2.2.2. Immune to Democratic Scrutiny

The public is not able to judge the social significance of biotechnology advances until scientists can declare with certainty what is realistic, until the imagined scenarios are already upon us. Although the controversy grew in breadth and intensity in anticipation [32,33], He Jiankui’s work violated a longstanding norm prohibiting genetic modification of the human germline [7]. This phenomenon not only occurs in the field of gene editing, but it is also relevant to the field of synthetic biology. In 2018, Canadian scientists obtained overlapping gene fragments through mail order, splicing them to synthesize a horse pox virus which was similar to the smallpox virus [34]. These artificially synthesized viruses are more capable of infecting, spreading, killing, and escaping than natural viruses, and therefore are more difficult to trace the origins of. The latter research was conducted without any knowledge of the public, and the public only discovers it subsequently. An academic survey of the US public on the acceptability of gene editing was conducted and most people chose to accept it, but the survey did not distinguish between somatic and heritable human gene editing, leading to considerable doubt as to whether most of the public demonstrated awareness of the difference between them [35]. A recent popular science book was written by one of the inventors of CRISPR/Cas9, and does not discuss heritable human gene editing, but simply advocates that gene editing can be used in humans [36].

Thus, the public’s role is one of dependence, with the public passively learning and deferring to science’s authoritative judgment about what is at stake and when a democratic reaction is warranted. More and more researchers have chosen to bypass the supervision of policymakers and the public, and directly implant their actions with germline and heritable human genome editing into major scientific decisions concerning the future of human society with internal discussion only. In 1975, Senator Ted Kennedy characterized Asilomar as a usurpation of democratic authority, saying, “They were making public policy, and they were making it in private” [37]. Correspondingly, after 18 scientists and ethicists from 7 countries called for a global moratorium on the clinical application of genetic editing in all human germ cell lines [38], some researchers currently conducting human germline genome editing research, including Shoukhrat Mitalipov [39] and Denis Rebrikov [40], openly acknowledged heritable genome editing as their ultimate aim [41]. For the public, those in the position to create the future of technology are also those with the most competence to declare what possible futures can warrant the public and to call a halt and break the rules. The risk of biotechnology is inherent, meanwhile, its internal structure biotechnology in-practice is immune to democratic scrutiny. This renders society and its institutions inevitably and perpetually reactive [42]. The scientists are the “boots on the ground” with respect to biotechnology, and they have the potential to be the best reporters of misuse, even if this failed in the He case [43].

## 3. Challenge of Governing Human Germline and Heritable Genome Editing

### 3.1. Ever-Lasting Strong Arguments against Engaging in Heritable Human Gene Editing

#### 3.1.1. Unjustified Medical Intervention

In the great majority of cases, heritable human genome editing is not required for genetic illnesses. For the general disease-causing genetic changes, preimplantation genetic testing and selection of embryos for implantation will be the safer and cheaper technological and social alternatives [44]. Currently, in clinical assisted reproductive practices, in vitro fertilization (IVF) is often used in combination with preimplantation genetic diagnosis (PGD) or preimplantation genetic screening (PGS) to reduce the number of unnecessary abortions, for example. The number of disease-causing genes in a fertilized egg determines whether the naturally occurring human germ cells should be sequenced and selected for gene editing. If a fertilized egg has only one or a few disease-causing genes, it is easy to select for implantation. Only in a few cases would there be medical benefits (in terms of avoiding genetic disease) that could not be obtained through preimplantation genetic diagnosis or through prenatal testing or (if desired) abortion. Clinically, it is rare for people to be homozygous for dominant diseases, which would be a couple that both have the same autosomal recessive disease at the same time. As a result, heritable human genome editing should be used only in exceptional circumstances, such as when there is a compelling reason to prevent major illnesses and no other way to select healthy oocytes for IVF. The prospective parents also need to consider if they want a genetically modified offspring and the potential risks for the offspring, and the possible social impacts both for them and for the society, particularly with the other alternatives available.

The greatest impetus for heritable human genome editing may have nothing to do with disease. Heritable human genome editing could be used to reduce the number of people born with so-called undesirable traits or increase the number born with improved and possibly new so-called desirable traits, which may happen eventually [10]. The possibility that human genome editing might be used for the enhancement of human traits is very controversial. For example, editing genotypes that do not exist in their offspring, night vision [45], and other ability enhancements. All of these are an excuse to go beyond the standard functional purpose of heritable human genome editing, and thus further promote gene editing affecting other non-disease traits in humans. This situation is to satisfy commercial interests rather than to meet medical treatment needs.

#### 3.1.2. Unpredictable Technology

Genome editing holds promise for correcting pathogenic mutations such as spinal muscular dystrophy, thalassemia, and retinal macular degeneration [46,47]. However, the link between genes and diseases is not as straightforward as it appears. Due to genetic pleiotropy being widely available at disease mutation locations [48], using gene editing to treat disease may cause complex side effects, putting patients at risk of suffering from other common diseases and even increasing mortality [49]. Particularly, off-target effects have always been “The Sword of Damocles” with gene editing technology for clinical treatment. These off-target effects are also of concern for the biosecurity community, as these unintended mutations can theoretically lead to worsening of the disease or fatal outcomes [50]. The off-target effects of gene editing can be subdivided into genomic and transcriptomic levels. Many studies have focused on the off-target effects of DNA at the genomic level [51,52].

Furthermore, single-base editing tools, which are the most advanced in this field (e.g., BE3 and ABE7. 10), also have mRNA off-target effects [53]. Frequently occurring RNA mutations can interfere with normal cell growth and potentially raise the risk of cancer [54,55,56]. CRISPR, a recently developed gene-editing tool, has become synonymous with rapid biological advancement. Nevertheless, the few off-target effects of CRISPR can have major impacts on an organism. These off-target effects are not exactly like side effects, because they are potentially more damaging [57]. For example, CRISPR could lead to potentially support a selective advantage to pre-existing oncogenic mutant cells (e.g., P53 or KRAS) [58], and chromosomal abnormalities, deletions, or fragmentation [59,60,61]. The potential short- and long-term harm from human embryo genome editing, including the potential consequences of genetic mosaicism, unintended off-target effects and, unwanted on-target effects, must be fully understood before options for such medical interventions are considered [10]. Heritable human gene editing should not be performed, regardless of if it is working for a few patients. More important is its impact on all of humanity.

#### 3.1.3. Foreseeable Population Hazard

From a technical standpoint, heritable human gene editing would be relatively straightforward, but technical feasibility, controllability, and clinical needs are never the only considerations for performing the trials. Heritable modifications resulting from human genome editing may pose greater safety and ethical issues than somatic human genome editing, due to its possible consequences for offspring and human society. Although people try their best to improve positive perceived traits, eliminate disease risk, or remove perceived negative traits from the future offspring, particularly by those with the means or access to editing and reproductive technology. Fears loom that if genome editing becomes acceptable in the clinic to stave off disease, it will inevitably come to be used to introduce, enhance, or eliminate traits for non-medical reasons. In addition, targeted changes to a person’s genome would be passed on for generations, through the germline (sperm and eggs), fueling fears that embryo editing could have lasting unintended consequences [62].

There is also a prospect of “consumer eugenics”, which are eugenics driven by parental choice rather than by state order, which would have similar results to traditional eugenics, such that ethicists are concerned that unequal access to such technologies could lead to genetic classism. The latest CRISPR-Cas9 research can produce male- or female-only litters with 100 percent efficiency [63], which has a potential risk to exacerbate the gender imbalance of the global population in the future if it is applied to humans. Meanwhile, it is common evolutionary knowledge that organisms, including humans, require a high level of genetic diversity. Because heritable human gene editing could counteract natural selection in populations and have unanticipated effects on the diversity of human variants in the gene pool, population biologists have proposed creating a bank of traits that have been screened out of populations in case they need to be reintroduced [3].

### 3.2. Uncontrolled Stakeholders’ Behavior

There is widespread agreement that biotechnology dangers are more dependent on user intent and capacity than risks from public health [64]. Generally, the cause of heritable human genome editing safety and ethical risk is uncontrolled stakeholder behavior under the comprehensive influence of both the internal and external governance. Moreover, when heritable human genome editing becomes safe, some scholars believe the somatic/germline barrier, which has long acted as an ethical restraint, will fall away [65]. As a result, resolving the old governance model’s shortcomings, particularly forgoing the legal rules, is a primary concern. The translation and application of heritable human genome editing in clinical medicine have thus created the complicated traits of internal autonomy and external governance.

#### 3.2.1. Unfunctional Self-Governance Model in Biotechnology under the Shadow of the Technical Characteristics

First, potential benefits of human genome editing include new strategies for diagnosis, treatment, and prevention of genetic disorders; new avenues to treat infertility; new ways to promote disease resistance; contribution to vaccine development; and enhanced knowledge of human biology [2]. However, diseases are subjectively defined by persons. For example, mental illness is currently diagnosed worldwide using the Diagnostic and Statistical Manual of Mental Disorders by the American Psychiatric Association. This manual is revised by its committee once every few years, sometimes with new definitions of illness and sometimes with the removal of previous definitions of illness. Ff there are objective indicators of the disease, the field of biotechnological research allows the definition of “health” and “disease” to be constantly adapted by doctors in the context of diagnosis and treatment, even if the scope of change is only between values, and new and unprecedented medical needs which are constantly emerging. From a biotechnological point of view, there is a strong sense of malleability with respect to socio-cultural, legal, ethical, and moral norms. For example, The International Society for Stem Cell Research relaxed the 14-day rule on culturing human embryos in its latest research guidelines [66].

Second, since the research and development environment and application scenario of human genome editing are concealed, anonymous, and opaque, policymakers and the public may not be able to acquire enough information with which they are affiliated. The resulting information asymmetry will cause difficulties in supervising the application of human genome editing in the early stages of disease diagnosis and treatment, in addition to drug research and development. In response to these information asymmetry risks, the field of biotechnology has always emphasized enhancing professional internal supervision mechanisms. Biotechnology researchers acquire their professional skills in a closed environment, at scientific conferences, under the guidance of institutional biosafety [67], ethics committees [68], and through peer review of papers submitted for publication. All of these have formed an enclosed internal supervision mechanism for research, training, and practice. These monitoring mechanisms determine that the final review decisions for monitoring risks and performing biosecurity risk assessments depend on the expert group from within the system, which may have some negative effects. For example, the internal standards and operational mechanisms within the scientific community may exclude the concerns, doubts, and interventions of the government and the public. The judicial system, as a third party with traditional punitive capability, needs to rely on the judgment standards and operational systems established by the life science community in resolving legal disputes caused by the misuse of human genome editing [69,70].

Third, for individual scientists working in biotechnology, recognition by their peers is more important. The comparative advantages that can be demonstrated over peers: career opportunities, industrial status, titles, research funding, salaries, etc., are often linked to the ability to achieve ‘technical advances’ to gain recognition by the peer group. Concurrently, this mechanism of “those who can get more” also tests the effectiveness of the work of individual scientists. For young scientists, losing the recognition of their technical advances from their peers may lead them to face dismissal and replacement. As a result, they may be compelled to test or even overstep regulatory borders to compete for technological advancements. For example, CRISPR significantly reduces the cost and expertise barriers of earlier gene editing methods. Timely regulatory updates are therefore necessary, as the demonstration effect of overstepping boundaries can be fatal for young scholars. Some researchers had successfully used gene editing in non-viable (triple nuclear) spare embryos from in vitro fertility treatments before He Jiankui [71,72].

#### 3.2.2. Unpredictable Risks in Biotechnology Leading to the Lagging behind of Traditional Governance Model

Policies and laws governing human genome editing and related technologies can be created by a variety of top-down public mechanisms [10], and the scope of these documents ranges widely, including laws enacted by national legislatures or other official lawmaking bodies, ministerial statements that have the force of law, administrative regulations (as in China), royal decrees (as in Saudi Arabia), and in a few countries, research ethics guidelines and ethics codes [41]. Some of these laws are broad, and human genome editing simply falls within their scopes. In other cases, laws are created specifically for this technology. While legal instruments are essential for creating penalties, statute law is harder to change than regulations and guidelines. With the advances of cutting-edge technology and the deepening understanding of its risks, these methods usually require a cycle of data collection, evaluation, and rule modification [73]. As the scientific community can only assess the biosafety, ethical, and moral issues of heritable human gene editing based on the assumption of potential risks, there is a dilemma that the legal system is not acknowledging the development of the technology. It will be difficult to predict and assess unintended long-term consequences of germline and heritable genome editing, such as effects that only occur later in life and results from the specific genetic background of an individual. Therefore, effective resolution of judicial disputes cannot be implemented due to the lack of necessary judicial determination of causality in the future. Meanwhile, the frequent requirement of such restrictive legislation on the intentionality on the part of the individual (men’s rea) has created a degree of policy uncertainty, particularly with respect to downstream restrictions on certain applications, such as clinical uses [74].

It must be noted that “laws and norms complement each other. Implementing norms through formal laws and regulations is a slow and arduous task, which often needs to overcome considerable resistance. Therefore, it is difficult to complete this work by enforcing laws” [75]. Some technical experts in this field believe that supervision by public authorities cannot keep up with the pace of technological change, and that bottom-up industry self-regulation is superior to the traditional “hard law” in with respect to cost, and therefore will be less likely to be resisted by practitioners [76]. Particularly with respect to technological development trends, community laboratories and DIY bio enthusiasts (also called “biohackers”) are using the technology in many cases to make biological science more accessible for those not in traditional science careers. There is potential for misuse. Though, currently, misuse is largely related to self-harm [77,78,79].

## 4. The Policy Updates of Heritable Genome Editing in China and the US

### 4.1. Regulatory Updates in China

In December 2019, a Chinese court in Shenzhen found He Jiankui and two others guilty of violating Article 336 of the Criminal Law of the People’s Republic of China, which prohibits engaging in medical activities without a license [7]. In China, Article 336 forms the basis on which He Jiankui was convicted. However, there are obvious gaps in the Chinese regulation system, particularly the guideline which can be rendered ineffective or inadequate in practice, and therefore needs to be mended. The Civil Code of the People’s Republic of China was adopted during the Third Session of the Thirteenth National People’s Congress on 28 May 2020. Article 1009 states: A medical and scientific research activity related to human genes, embryos, or the like, shall be done by the relevant provisions of laws, administrative regulations, and the regulations of the State, and shall not endanger human health, offend ethics and morals, or harm public interests [80]. This is the first time that medical and scientific research activities related to human genes and human embryos have been clearly defined in relation to legal trial in China. The Amendment (XI) to the Criminal Law of the People’s Republic of China, as adopted at the 24th Session of the Standing Committee of the Thirteenth National People’s Congress of the People’s Republic of China, came into force on 1 March 2021. One article was added after Article 336 of the Criminal Law as Article 336B: “Whoever implants any genetically edited or cloned human embryo into the body of a human being or animal or implants any genetically edited or cloned animal embryo into the body of a human being shall, if the circumstances are serious, be sentenced to imprisonment of not more than three years or limited incarceration and a fine or be sentenced to a fine only; or if the circumstances are particularly serious, be sentenced to imprisonment of not less than three years nor more than seven years and a fine” [81]. However, there is currently no definition of “serious” in Article 336B of the Criminal Law. Currently, China adopts restrictive policy approaches, including laws and regulations, that outlaw implants of any genetically edited or cloned human embryo into the body of a human being or animal regardless of its purpose using tight regulations, blanket prohibitions, or moratoria.

From the recent advances in genetic engineering legislation, China is adhering to the precautionary principle as it believes that uncertainty in science and technology should not become a cause for the delay in adopting measures to prevent harms or threats, arguing that whoever carries on the development of biotechnology is obliged to bear the burden of proof of no harm. Should any biotechnological achievement carry risks on which no scientific consensus has been gained, there would be a need to impose oversight for prevention and precaution. This approach is analogous to the EU’s “safe enough” framework for genome editing, in which the unrestricted deployment of a technology is conditioned on achieving a certain level of safety. However, it is worth mentioning that the EU has begun to explore the inadequacies of the “safe enough” framework, such as “how safe is safe enough” and the confinement of ethical and governance reflections to safety issues [65].

### 4.2. Regulatory Updates in the US

The United States does not have explicit legislation in place permitting or banning work in human embryos outright, considering such research experimental and not therapeutic, but has accepted the 14-day rule barring research on embryos after they reach a key point of complexity as a standard that guides researchers, reviewers, and regulators. After the He Jiankui affair, the Further Consolidated Appropriations Act of 2020 contains one provision that restricts federal funding of human embryo research and another that prohibits the Food and Drug Administration from considering applications for clinical trials involving heritable human genome editing [82]. Furthermore, although human germline genome editing is prohibited with the use of federal funding but is not otherwise prohibited, some US states do have such restrictions [83], but rules in other states are less clear and some do not ban it at all. In these states, researchers could do research on embryos with private funding. Additionally, researchers who want to investigate the clinical uses of genetically engineered somatic cells must secure people’s informed consent, and this occurs under the oversight of the Food and Drug Administration and the Department of Health and Human Services in the US [66].

In general, the US takes proactive approaches to the regulation of human germline genome editing based on the experience collected in practices in biotechnology-related fields to frame technological standards of assessment that involve public policies on biosafety and ethics. The US claims that it will not release policies for more stringent oversight of advanced biotechnology until biosafety risks have been established because it views intellectual freedom for innovation to be crucial, even if biosafety risks must be anticipated [84]. The preceding US perspective is consistent with Max More’s “proactionary principle”. According to more, the proactionary principle urges all parties to actively take into account all the consequences of an activity-good as well as bad- while apportioning precautionary measures to the real threats we face. In addition, to do all this while appreciating the crucial role played by technological innovation and humanity’s evolving ability to adapt to and remedy any undesirable side-effects” [85]. Therefore, this principle advocates the anticipation and assessment of the adverse effects that technological developments may reveal, and more importantly, emphasizes the need to learn about those effects through action.

## 5. Toward Inclusive Global Governance on Experimentalist Governance Model

The current and potential risks of germline and heritable human genome editing research will go beyond national borders, as will possible societal effects. The competition in applying this technology is unbelievably fierce, including for commercial purposes. For example, there was a strong debate about the competition between countries in developing/using CRISPR [86]. Therefore, governance for this technology is needed at national levels and transnational levels. We appreciate the new framework developed by the WHO committee, which highlights the role of various tools, institutions, processes, and staff in the governance of human genome editing [1]. It is necessary to establish one or more multisectoral, transdisciplinary, and collaborative governance mechanisms promoted to more effectively address the public health and social threats in applying this technology. The key to achieving these goals is more democratic deliberation of the public and the inclusive engagement of the global scientific community.

### 5.1. Involving Democratic Deliberation

First, as a democratic deficit, consigning the public to a reactive role renders democracy subordinate to epistemic correctness. It reflects that the public was limited to the mere reaction by uncritically deferring to scientific accounts of what scenarios are realistic and what public reactions are warranted [87]. Being honest with the public about the advantages and drawbacks of reproductive systems and heritable human genome editing would be critical. This challenge cannot only be rectified by cultivating a more informed and engaged public, but also needs to implement the principle of democratic deliberation. If the conclusions are unlikely to satisfy all stakeholders, democratic deliberation encourages respectful debate of opposing viewpoints and active participation by all stakeholders as a method of collaborative decision-making. Careful public dialogue and debate with open interchange among all stakeholders will foster a constructive atmosphere and can promote the perceived legitimacy of outcomes by the public [88].

Second, delivering the democratic deliberation implies that not only should different expertise and nations be represented in an ongoing multidisciplinary dialogue, but also the lay public who must be informed to support transparency should also be included. To ensure a participatory process during the discussion, democratic deliberation should be engaging a wide range of perspectives and expertise and be consulted widely, and solicit information about societal attitudes towards several areas of research and clinical use of human genome editing technology. This includes biomedical scientists, social scientists, ethicists, health care providers, patients and their families, people with disabilities, policymakers, regulators, research funders, faith leaders, public interest advocates, industry representatives, and members of the general public [89]. In-person and online topic-specific meetings are necessary to consult with individuals and representatives of organizations including international agencies, academies of science and medicine in addition to other national or professional bodies, patient groups, and civil society organizations. In addition, one important barrier among a diverse group of stakeholders’ multidisciplinary discussion is the (potential) lack of knowledge and/or understanding of different publics [90]. To support autonomy and informed decisions, all stakeholders involved should be appropriately educated about the basic science and possibilities of human gene editing.

Third, balancing therapeutic prospects brought by scientific advances with regulation to address highly contested socio-ethical issues, public deliberation on human genome editing should transcend the technologies that make it urgent. Heritable human genome editing repeatedly oversteps boundaries and exposes the tension between the science community and public deliberation. The crux of the problem lies not only in biotechnology, but it is also closely related to the situation that biotechnology itself is subordinate to public deliberation of law and ethics. We should not leave it to scientific experts alone to determine when a moratorium is necessary or that governance needs to play catch-up. There will inevitably be a need to form a harmonious coexistence and cooperative advance between biotechnology, law, ethical, and moral considerations under the ultimate challenge of dealing with disruptive science. Among these, ethical and moral considerations are usually in the form of soft laws and informal measures, such as professional standards, codes of ethics, and education, in addition to awareness measures [91]. Meanwhile, cultivating and establishing a responsible culture through moral education in the life science community may be more valuable and practical [92]. Additionally, surveys indicate that many life science researchers still lack awareness of the misuse and abuse risks associated with their research [93]. To ensure that researchers can truly understand and implement the codes of conduct, institutions must ensure that their members recognize and accept the responsibilities and consequences of violating the codes based on understanding and learning [94]. The education and training as part of outcomes in democratic deliberation could suggest actions for those with concerns and provide information about where scientists should turn if they have concerns.

### 5.2. Toward Inclusive Global Governance on Experimentalist Governance Model

Such biomedical and regulatory situations urge each government to reconsider the current policies on germline gene modification. Notably, heritable human gene editing is not prohibited worldwide. Some countries have signed an international convention (e.g., Oviedo Convention [95]) while others outside of the regime have additional policy documents relating to gene editing in this area. This technology will inevitably be misused in some countries with lax regulations that prohibit such procedures under guidelines that involve less enforcement than the law. We need to draw lessons from broader global governance practices and recommend experimentalist governance for charting a scalable path forward, as the goal of experimentalist governance is to reflect on and institutionalize policies in grassroots practice according to local conditions [96]. This process includes setting framework goals, implementations target for each country, regular reporting, the conduct of peer evaluation, and periodically revising the framework [97].

First, effective governance requires recursive learning in decentralized compliance with international conventions, including governance frameworks at transnational levels by various types of global rule makers, such as intergovernmental organizations (e.g., UNESCO and the WHO Expert Advisory Committee on Developing Global Standards for Governance and Oversight of Human Genome Editing), international non-governmental organizations (e.g., The Hinxton Group, Baltimore, MD, USA), national scientific committees (e.g., the Nuffield Council on Bioethics and Royal Society of the UK; the National Academies of Sciences, Engineering, and Medicine of the US), international scientific conference committees (e.g., the Organizing Committee of the Second International Summit on Human Genome Editing), and some parallel statements published in leading academic journals (e.g., Nature [38]). Some of the researchers who created gene-editing technology called for prohibiting both clinical applications of these technologies and the forms of research that would make such applications possible [38]. By contrast, the U.S. National Academy of Medicine, the U.S. National Academy of Sciences, and the U.K.’s Royal Society considered potential clinical applications of human germline genome editing and evaluated if society should conclude that heritable human genome editing applications are acceptable [98]. Despite becoming more socially aware, equitable, and just, these different recommendations and statements do not entirely concur with one another. It is crucial to achieve a consensus on a general global governance framework with open-ended principles, which is a choice between prohibition and acquiescence to an established technological trajectory. With biotechnology research and application continuing to progress faster at the global level, it becomes necessary to ensure international coordination of governance measures through or under the auspices of the United Nations or other multilateral agencies [99]. Idealistic venues would be the United Nations Educational, Scientific and Cultural Organization (UNESCO) or the Biological Weapons Convention, the former considering Article 16 of the Universal Declaration on Bioethics and Human Rights (2003), the latter due to its the most powerful binding force under the current framework of international biotechnology security governance. The common solution could be a new declaration or an addendum that considers gene editing, one that would bind the states’ parties.

Second, each country’s practice considering local conditions should guide the country through policy uncertainties. One problem that can be observed in the above frameworks is the vagueness of the regime concerning the issue of use at the institutional, national, regional, and international levels, in addition to being sustainable. Eventually, differences in national policy might start to reemerge throughout the world. Although it might be difficult to impose worldwide legislation standards, each country should be encouraged to adopt the general standards to its circumstance. Creating a culture of responsibility in the human gene editing community could do more to promote responsible stewardship in human genome editing than any other single strategy. The culture prioritizes facilitating effective biosafety and biosecurity awareness-raising in life science research, encouraging research integrity and responsible use of science, improving practitioner accountability, and fostering access to developing capabilities [100]. There are actors in the world of human genome editing, namely community laboratories and Do-It-Yourself (DIY) bio enthusiasts (also called “biohackers”), and others, who practice outside of conventional biological or medical research settings. These groups may not be familiar with the standards for ethics and responsible stewardship that are commonplace for those working in biomedical research. This poses a new challenge regarding the need to educate and inform human gene editing researchers in all communities about their obligations, particularly regarding biosafety, ethics and morals. Such education is also important in countries with legislation on germline gene modification because it could have unintended consequences, including “genome engineering tourism” to lax sovereignties. There may be some parallels with discussions about legislation for abortion and euthanasia in this regard.

Third, recognizing that international coordination is essential, and governments should act to ensure an ongoing dynamic consultation mechanism for transnational information sharing and action review. A consensus on a general global governance framework could make a guide for the questions and provide a basis for each country to formulate its oversight policies according to its realities and cultural, political, religious, and social context. A mechanism or body should be identified to (1) leverage existing resources by providing an ongoing and coordinated review of developments in human gene editing, (2) ensure that regulatory requirements are consistent and non-contradictory, and (3) periodically and in a timely basis inform the public and international governments, the World Health Organization, and other appropriate parties of their findings. Governments should continue and expand efforts to collaborate although political relations are otherwise deteriorating.

Fourth, good governance is an iterative, ongoing process that includes mechanisms for regular revision. Ideally, it is proactive, not only reactive. Discussions of safety, ethics, and moral objections to human gene editing should be revisited periodically as research in the field advances in novel directions. Reassessment of concerns regarding the implications of germline and heritable human genome editing for humans and society should track the ongoing development of the field. This reassessment should not take longer than one year and should consider scientific, technological, and societal changes, adequacy of implementation and assessment of impact, and potential future needs or concerns. Based on the three stages above, experimentalist governance will ultimately achieve the renewal and iteration of the global governance mechanism.

We believe that worldwide cooperation on the governance of heritable human genome editing would be beneficial, as this may be the unique period for the policy to mature development. The COVID-19 pandemic has demonstrated the importance of using new tools and methods to combat serious diseases and has highlighted the potential biosafety risks of human genome editing research, both of which are crucial in forging a consensus on global governance. The benefits in addition to risks and negative impacts encountered when conducting heritable human gene editing in any research context should be adequately monitored, and information about these should be made readily available. In the second half of 2022, the States Parties to the Biological Weapons Convention will hold their Ninth Review Conference. Notably, taking advantage of the close ties with biological scientists, the Implementation Support Unit (ISU) of BTWC organizes, co-organizes, and participates in major international conferences and other activities in the life science research field to successfully improve scientists’ awareness of the potential risks of abuse and intentional injury, and promote scientist groups to participate in the implementation of responsible science behaviors. To address the effective implementation of advocating responsible life science research at the global level, practicing heritable human genome editing under the framework of the BTWC seems to be a good choice. We call on the world to take advantage of this opportunity and allow this biotechnology to benefit global development.

## 6. Conclusions

Amid a plethora of debates over the governance of heritable human gene editing, it is crucial to focus on reasons for stakeholder non-compliance with norms and expectations in this area. Scientific and technological developments in gene editing are developing rapidly, and governance remains far behind.

A fresh look at the predicaments of governance on germline engineering argues for a two-part response: both the creation of laws in addition to measures with respect to self-governance. The key to achieving these goals is updating regulations and improving the awareness or education of responsible conduct. Therefore, for the policymaker, the most efficient way to be prepared for prohibitive policies of heritable human gene editing is in the laws, which should be enforced by the legislation, and the latter should influence regulation and guidelines and/or the articulation of relevant codes of conduct, etc. Particularly, cultivating and establishing a responsible culture through moral education in the life science community may be significantly valuable and practical. To this end, China and the US are two good examples of what should be done, and have implications for other countries. To address this issue, both states have promulgated legally binding measures, regulatory and/or professional guidance, and to address violations, the former and latter are accompanied by criminal sanctions with imprisonment. In addition, to ensure the regulatory effect of heritable human gene editing and promote a culture of responsibility and guard against such misuse, scientists from China and the US also jointly developed the Tianjin Biosecurity Guidelines for Codes of Conduct for Scientists [101]. Improving governance of heritable human gene editing worldwide and beyond could be achieved, we argue, under the cooperation between the WHO and BTWC, the former considering the new framework for the governance of human genome editing, and the latter because it successfully improved scientists’ awareness of the potential risks of abuse and intentional injury, which can be enhanced through the various biological security education program [102]. Together, implementation of the above measures can promote the better engagement of scientist groups with this issue and improvements in responsible science behaviors.

## Data Availability

Not applicable.

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
