# Peer review of "Governance of Heritable Human Gene Editing World-Wide and Beyond"

_ijerph, 2022, doi:10.3390/ijerph19116739_

Round 1

Reviewer 1 Report

Novel genome editing methods have opened new dimensions on the ethical evaluations concerning the risks and consequences of using these techniques of changing genes in treatments of human diseases and reproduction. The main concerns are the lack of universal guidelines for implementation of these techniques by the scientific community and the acceptance by the public, as they involve great ethical challenges.

The manuscript provides an impressive systematic overview and discussion on the topic of human gene editing. The authors distinguish between human genome editing for clinical application that involves somatic (non-heritable) cells and germline genome editing, which involves heritable genetic changes. While the application of these techniques for human therapy and prevention of disease is bound by existing policies and ethical norms which focus on patients’ safety and efficacy, germline genome editing has been universally viewed as unacceptable at the present time. However, there are ongoing debates and deliberations by the scientific community on the barriers and the approach that should be taken in the future. The main effort should be in arousing public awareness for establishing robust and credible regulations that will be offered by the scientific community, taking into account a broad societal set of values or alternatively, present strong arguments against the use of this technology.

Most of the manuscript is devoted to the approaches for improving the interaction between scientists and the public and the increasing public demands for the right be informed and right to convey contradictory opinions and not leave decisions in the hands of scientists.

The manuscript presents well the arguments against engaging in heritable genome editing, from the limited advantage in selection of genetically modified children free of a genetic disease, the unpredictable outcome for correcting pathogenic mutations, the fear for unintended consequences to changes affecting the human population. Moreover, the authors highlight the limitations of current regulations, the closed environment whereby scientific research is practiced, with limited consideration of public or governmental concerns and of the dangers of individual scientists to gain recognitions by their peers by disregarding boundaries.

The authors mention that the majority of countries prohibit heritable human genome editing and the policies vary between them. They advocate for a global approach and describe in detail the laws and regulations in China and the USA as a basis for future deliberations on updating and improving guidelines, regulations and legislation.  

Overall, this is a very informative and detailed review presenting novel approaches to the topic of regulation and legislation concerning heritable human genome editing.  

Suggestions:

  1. While there has been a consensus that a moral divide exists between somatic and germline human gene editing, according to Evans (PNAS 2021, 118 no. 22 e2004837117) this barrier has been weakened. This view should be also presented in the manuscript.
  2. In addition to the legislature in China and the USA, the authors should attempt to present the view of the European Community, as they aim for world-wide governance. One publication on these lines is: Ethics of genome editing, https://ec.europa.eu/info/sites/default/files/research_and_innovation/ege/ege_ethics_of_genome_editing-opinion_publication.pdf

Author Response

We thank you for the comments on our paper and the valuable and constructive suggestions. We have included that information in the revised manuscript. For suggestion 1, please see lines 285 – 286, and suggestion 2 on lines 411-416.

We also made a substantial cut on the whole manuscript requested by the editors. Therefore we include the clean edited file for your information.

Reviewer 2 Report

GENERAL COMMENTS

This very long manuscript contains a very credible suggestion to take advantage of the expertise and cachet of two international organizations (WHO and TBWC) to oversee ethical aspects of human germline gene editing. The feasiblity, pros, and cons of the suggested collaboration need to be spelled out.  The comparison of China and US procedures is helpful. I admire the summary sentence buried in lines 665-667.

SPECIFIC COMMENTS (with line numbers)

 19       Experimentalists governance is unclear

 22       Under the cooperation does not make sense

 30       Most fast not appropriate

48-34   Use commas, not semicolons

39        Small h in human

42-46   Garbled sentence

47        Could, not would

51        Sentence seems duplicative

53        That should be deleted

56        Human gene name abbreviations must be in italics; virus is duplicative

67        It's probably good to insert human before germline

70        ? should be a colon

71        Insert upon, after agreed

73        Is "engagement of this" wrong?

98        ? Seems wrong

104-112  Far too long a sentence

117      Deepening of the modernization does not make sense

125      Meaning of "this: is unclear

130      ...considerations. [15] but....  is not clear

146      Needs citation for public believe

149      Undouble not a word

150      To recognize is awkward

156      "Unprecedented informed consent" is unclear

160-163.  you can Very confused sentence

163      what does "It" refer to?

231`     "Selected or gene editing" is confusing

237      Rather rare needed is confusing

257f     Opening clause is confusing  

268      Human gene names must be capitalized; longer duration of mutations is confusing

275      Small c in chromosomal

284      Trials is misspelled

302      Citations needed for both sentences

307,311 Out-controlled is not a word

310      More than what?

314      Have, not has

315      As follow is unclear

339      Field.. has; or fields...have

349      System...needs; or systems...need.

352,358  Not from within, but by their peers 

359      Expel them to test is unclear

369      Small t and d in top-down

389      Implementing....

403      Night-six is not a number

404      Prohibiting

406      Policy makers 

413-416 Awkward

478-480 Definitions of serious are needed

498f     Pick U.S. or US, but be consistent

529-532 Somewhat garbled

546-548 Extraneous comment ?by prior reviewed, to be deleted

559-561 Not a sentence

610      while stick is unclear

618      all lowercase H

620      need reference to the Oviedo Convention

638      probably should journal names and give exact citations

668      security dimensions is unclear

672      DIY it should be spelled out

705      what is meant by the window of policy opportunities?

712      will be held is wrong

734      suggestions on education this would be welcomed

742      do you need a citation for the Tianjin guidelines?

Author Response

We thank you for the valuable and constructive comments and suggestions on our paper, especially the careful scrutiny of the details of the manuscript.  We now have made changes accordingly and listed as below.

We also made a substantial cut on the whole manuscript requested by the editors. Therefore we include the clean edited file for your information.

SPECIFIC COMMENTS (with line numbers)

Comments 1:  19      Experimentalists governance is unclear

Response 1: We have added the explanation of the “Experimentalists governance” in lines 19-20 which is a multi-layered model of governance that establishes an open-ended framework from the top and offers stakeholders the freedom of discussion.

Comments 2:   22       Under the cooperation does not make sense

Response 2: We have changed the sentence to “we believe that practicing heritable human genome editing in accordance with the WHO and BTWC appears to be a good choice in lines 23-24.

Comments 3:   30       Most fast not appropriate

Response 3: We have changed it to “most rapidly emerging” in line 31.

Comments 4: 48-34   Use commas, not semicolons

Response 4: We have changed this semicolon to a comma in line 35.

Comments 5:  39        Small h in human

Response 5: We have changed it to h in human in line 37.

Comments 6: 42-46   Garbled sentence

Response 6: We have deleted the garbled sentences.

Comments 7:  47        Could, not would

Response 7: We have changed this “would” to “could” in line 40.

Comments 8:  51        Sentence seems duplicative

Response 8: We have changed the duplicative sentence to “Before Heritable germline editing can fulfill the proper risk and benefit standards for clinical trials, much more research is required” in line 43-44.

Comments 9: 53        That should be deleted

Response 9: We have deleted the word “that” in line 44.

Comments 10:  56        Human gene name abbreviations must be in italics; virus is duplicative

Response 10: We have changed the CCR5 to CCR5 and deleted the virus in line 48.

Comments 11:  67        It's probably good to insert human before germline

Response 11: We have inserted “human” before “germline” in line 59.

Comments 12:  70        ? should be a colon

Response 12: We have changed the “?” to a colon in line 62.

Comments 13:  71        Insert upon, after agreed

Response 13: We have inserted “upon” after “agreed” in line 64.

Comments 14:  73        Is "engagement of this" wrong?

Response 14: We have changed the sentence to “…, particularly in those technological engagements where unjustified, unpredictable, and foreseeable characteristics bring strong arguments” in line 65.

Comments 15:  98        ? Seems wrong

Response 15: We have changed the “?” to a comma in line 90.

Comments 16:  104-112  Far too long a sentence

Response 16: We have deleted that long sentence.

Comments 17:  117      Deepening of the modernization does not make sense

Response 17: We have changed the sentence to “With the advancement of science and technology…” in line 95.

Comments 18:  125      Meaning of "this: is unclear

Response 18: We have changed the sentence to “Although the action above is indeed due to scientists' professional ethics, it has virtually intensified the disintegration of their professional authority and status” in line 103.

Comments 19:  130      ...considerations. [15] but....  is not clear

Response 19: We have changed the sentence to “It not only demonstrates public dissatisfaction with current policy considerations, [15] but also expresses public and scientific different viewpoints on whether such heritable genome editing is needed or not” in lines 107-108.

Comments 20:  146      Needs citation for public believe

Response 20: We have added a new citation in the 126th line.

Comments 21:  149      Undouble not a word

Response 21: We have deleted the “Undouble” in line 126.

Comments 22:  150      To recognize is awkward

Response 22: We have changed the sentence to “These changes are positive and worth to be recognized” in line 127.

Comments 23:  156      "Unprecedented informed consent" is unclear

Response 23: We have changed the "Unprecedented informed consent" to “informed consent” in line 133.

Comments 24:  160-163.  you can Very confused sentence

Response 24: We have changed the sentence to “This is because biotechnology's original goal of ruling nature has shifted to technological dominance over human subjects, regardless of whether the reality has resulted in a significant infringement or is just speculation about the future” in lines 137-139.

Comments 25:  163      what does "It" refer to?

Response 25: We have changed the sentence to “As a result of the aforementioned changes, biotechnology may face resistance from social culture, ethics, and morality” in lines 139 -140.

Comments 26:  231`     "Selected or gene editing" is confusing

Response 26: We have changed the “or” to “for” in line 206.

Comments 27:  237      Rather rare needed is confusing

Response 27: We have changed the sentence to “As a result, heritable human genome editing should be used only in exceptional circumstances, such as when there is a compelling reason to prevent major illnesses and no other way to select healthy oocytes for IVF” in lines 211 -213.

Comments 28: 257f     Opening clause is confusing  

Response 28: We have changed the sentence to “Frequently occurring RNA mutations can interfere with normal cell growth and potentially raise the risk of cancer” in lines 242-243.

Comments 29:  268      Human gene names must be capitalized; longer duration of mutations is confusing

Response 29: The human gene name “P53” has been capitalized, and we have changed the sentence to “Frequently occurring RNA mutations can interfere with normal cell growth and potentially raise the risk of cancer” in line 242.

Comments 30:  275      Small c in chromosomal

Response 30: We have changed it in line 249.

Comments 31:  284      Trials is misspelled

Response 31: We have changed it to “trials” in line 258.

Comments 32:  302      Citations needed for both sentences

Response 32: We have added a new citation in line 279.

Comments 33:  307,311 Out-controlled is not a word

Response 33: We have changed it to “Uncontrolled” in lines 280 and 283.

Comments 34:  310      More than what?

Response 34: We have changed the sentence to “There is widespread agreement that biotechnology dangers are more dependent on user intent and capacity than risks from public health” in line 282.

Comments 35:  314      Have, not has

Response 35: We have changed the “has” to “have” in line 289.

Comments 36:  315      As follow is unclear

Response 36: We have changed the sentence to “To this purpose, the translation and application of heritable human genome editing in clinical medicine have created the complicated traits of internal autonomy and external governance” in lines 288- 290.

Comments 37: 339      Field.. has; or fields...have

Response 37: We have changed the “fields” to “fields” in line 314.

Comments 38:  349      System...needs; or systems...need.

Response 38: We have changed the “need” to “needs” in line 325.

Comments 39:  352,358  Not from within, but by their peers 

Response 39: We have replaced the “within” with the “by” in lines 328 and 331.

Comments 40: 359      Expel them to test is unclear

Response 40: We have changed the sentence to “As a result, they may be compelled to test or even overstep regulatory borders to compete for technological advancements” in line 335.

Comments 41: 369      Small t and d in top-down

Response 41: We have changed the word to “top-down” in line 345.

Comments 42: 389      Implementing....

Response 42: We have changed the “Implement” to “Implementing” in line 365.

Comments 43: 403      Night-six is not a number

Response 43: According to the length of this article, we have deleted this section.

Comments 44: 404      Prohibiting

Response 44: According to the length of this article, we have deleted this section.

Comments 45: 406      Policy makers 

Response 45: According to the length of this article, we have deleted this section.

Comments 46: 413-416 Awkward

Response 46: According to the length of this article, we have deleted this section.

Comments 47: 478-480 Definitions of serious are needed

Response 47: We have added the sentence “However, there is currently no definition of “serious” in Article 336B of the Criminal Law” to illustrate the problem in line 400.

Comments 48: 498f     Pick U.S. or US, but be consistent

Response 48: We have changed the “U.S.” to “US” in line 417.

Comments 49: 529-532 Somewhat garbled

Response 49: We have changed the sentence to “The US claims that it will not release policies for more stringent oversight of advanced biotechnology until biosafety risks have been established because it views intellectual freedom for innovation to be crucial as well, even if biosafety risks must be anticipated” in lines 436 – 439.

Comments 50: 546-548 Extraneous comment ?by prior reviewed, to be deleted

Response 50: We have added the corresponding content in lines 463 - 474.

Comments 51: 559-561 Not a sentence

Response 51: We have changed the sentence to “Even if the conclusions are unlikely to satisfy all stakeholders, democratic deliberation encourages respectful debate of opposing viewpoints and active participation by all stakeholders as a method of collaborative decision-making.” in lines 470 – 472.

Comments 52: 610      while stick is unclear

Response 52: According to the length of this article, we have deleted this section.

Comments 53: 618      all lowercase H

Response 53: We have changed them in line 516.

Comments 54: 620      need reference to the Oviedo Convention

Response 54: We have added a new citation in line 518.

Comments 55: 638      probably should journal names and give exact citations

Response 55: We have added a new citation in line 538.

Comments 56: 668      security dimensions is unclear

Response 56: We have changed the sentence to “The culture prioritizes facilitating effective biosafety and biosecurity awareness-raising in life science research, encouraging research integrity and responsible use of science, improving practitioner accountability, and fostering access to developing capabilities” in lines 566 – 569.

Comments 57: 672      DIY it should be spelled out

Response 57: We have added “Do-It-Yourself” before “DIY” in line 570.

Comments 58: 705      what is meant by the window of policy opportunities?

Response 58: We have changed the sentence to “We believed that worldwide cooperation on the governance of heritable human genome editing would be beneficial, as this may be the unique period for the policy to mature development” in lines 603 – 604.

Comments 59: 712      will be held is wrong

Response 59: We have changed the sentence to “In the second half of 2022, the States Parties to the Biological Weapons Convention will hold their Ninth Review Conference” in lines 610 – 612.

Comments 60: 734      suggestions on education this would be welcomed

Response 60: We have changed the sentence to “…, which for example can be enhanced through the various biological security education program” in line 647.

Comments 61: 742      do you need a citation for the Tianjin guidelines?

Response 61: We have added a new citation in line 648.

Reviewer 3 Report

In the publication under the title Governance of Heritable Human Gene Editing World-Wide and Beyond, which has been made available to me for review, the authors give a thorough overview of the law and application of gene editing technology in modern medicine. The authors point out  possible applications of this technique, which can be used for the genome of somatic cells as well as for the genome of germ cells. The authors also point out the potential contribution of this method as a new diagnostic tool. I agree with the authors that there is an urgent need to develop international standards for the use of gene editing in the human genome, especially when it is used as a therapeutic element. It is also necessary to point out the great dangers involved in trying to manipulate the human genome at the level of embryonic development. This is a completely unacceptable method, not only for ethical reasons, but also because of the unpredictability of the process on the growing organism.
This publication is also a thorough, multidirectional analysis of the risks associated with the
use of gene editing techniques in medicine. This publication makes an important contribution to the review of the positions of government institutions that support the use of this technique in the research and treatment process. It considers that it should be published in its present form.

Author Response

We very much appreciated the reviewer’s comments and the straightforward recommendations for publishing the paper as the current form.